# The Impact of Emotional Symptoms and Family Support on the Association Between Homophobic Bullying and Sedative/Hypnotic Use among Gay and Bisexual Men in Taiwan: A Moderated Mediation Model

**DOI:** 10.3390/ijerph17113870

**Published:** 2020-05-29

**Authors:** Dian-Jeng Li, Yu-Ping Chang, Yi-Lung Chen, Cheng-Fang Yen

**Affiliations:** 1Graduate Institute of Medicine, College of Medicine, Kaohsiung Medical University, 100 Tzyou 1st Road, Kaohsiung 80708, Taiwan; u108800004@kmu.edu.tw; 2Department of Addiction Science, Kaohsiung Municipal Kai-Syuan Psychiatric Hospital, 130, Kaisyuan 2nd Road, Lingya District., Kaohsiung City 80276, Taiwan; 3School of Nursing, The State University of New York, University at Buffalo, 12 Capen Hall, Buffalo, New York, NY 14260-1660, USA; yc73@buffalo.edu; 4Department of Healthcare Administration, Asia University, 500, Liufeng Road Wufeng District, Taichung 41354, Taiwan; 5Department of Psychology, Asia University, 500, Liufeng Road Wufeng District, Taichung 41354, Taiwan; 6Department of Psychiatry, Kaohsiung Medical University Hospital, 100 Tzyou 1st Road, Kaohsiung 80708, Taiwan

**Keywords:** sexual minority, sedative/hypnotic, family support, emotional symptoms, homophobic bullying

## Abstract

Sedative/hypnotic use and homophobic bullying have become a big mental health concern for gay and bisexual men. However, few studies have investigated the mediators and moderators of the association between them. The current study aimed to build a conceptual model to estimate the mediating effect of emotional symptoms and the moderating effect of family support on this association among gay and bisexual men in Taiwan. A total of 500 gay or bisexual men were recruited for the study. Their history of homophobic bullying, their experience of sedative/hypnotic use, their perceived family support, and their current emotional symptoms were evaluated using self-reporting questionnaires. A moderated mediation model was developed to test the mediating effect of emotional symptoms and the moderating effect of family support. A higher level of homophobic bullying was significantly associated with sedative/hypnotic use among gay and bisexual men and this was mediated by a higher severity of emotional symptoms. A moderating effect of family support was identified, wherein the mediating effect of emotional symptoms was weaker when there was a higher level of perceived family support, thus revealing the protective effect of family support. The significant impact of emotional symptoms and family support on the association between homophobic bullying and sedative/hypnotic use was identified. Timely interventions for emotional symptoms and the enhancement of family support are crucial for gay and bisexual men.

## 1. Introduction

### 1.1. Sedative/Hypnotic Use in Gay and Bisexual Men

Problematic substance use has become a significant mental health burden for gay and bisexual men [1,2]. The abuse of prescription drugs, including sedatives/hypnotics, has become a key health concern. Benzodiazepine binds to γ-aminobutyric acid (GABA)/barbiturate receptors causing muscle relaxing effects [3], and it interacts with αGABA_A_ subunits leading to sedative and anxiolytic reactions [4]. Benzodiazepine has also been reported to have a weak reinforcing effect, which leads to abuse [5]. One cohort study, where 90.2% of the participants were gay and bisexual men, reported a 10.2% rate of self-reported use of benzodiazepines in the past six months [6]. Another cross-sectional study in Taiwan reported that 5.4% of gay and bisexual men used sedative/hypnotic drugs in the preceding month [7]. Studies regarding the misuse of sedative/hypnotic drugs and their risk factors are warranted in gay and bisexual men.

### 1.2. Effect of Homophobic Bullying and Victimization on Sedative/Hypnotic Use

Individuals from the group of gay and bisexual men often suffer from homophobic bullying. A cross-sectional study demonstrated that gay men reported more frequent bullying and homophobic bullying than heterosexual men [8]. Homophobic bullying also causes harmful effects on the mental health of gay and bisexual individuals. It was reported to be associated with internalized sexual stigma and violation of traditional gender roles [9]. Moreover, a Taiwanese study reported that victimization of homophobic bullying was significantly related to self-identity confusion in gay and bisexual men [10]. Few studies have explored the hazardous effects of homophobic bullying and victimization on substance abuse for sexual minority individuals. Both experiencing bullying and being part of a sexual minority are risk factors for the misuse of prescription drugs [11]. However, factors that may mediate or moderate the association between homophobic bullying and sedative/hypnotic use have yet to be investigated.

### 1.3. The Role of Emotional Symptoms and Family Support

Substantial evidence suggests that sexual minority youths experience more severe depression compared with their heterosexual peers [12,13]. A previous epidemiological study indicated that about one third of young men (90% of them were gay or bisexual men) with substance abuse problems had attempted suicide, and more than half of them had a high level of depressive symptoms [14]. A recent review confirmed that bullying is one of the predominant risk factors for depression among sexual minority youths [15]. The high rate of depression in sexual minority youths was contributed to by discrimination and bullying [16]. However, the manner in which emotional symptoms, such as depression and anxiety, are involved in the interaction between bullying victimization and substance use remains unclear. The mediating effect of emotional symptoms on the association between homophobic bullying and sedative/hypnotic use in gay and bisexual individuals has not been well investigated. If emotional symptoms mediate the association between homophobic bullying and sedative/hypnotic use, intervention strategies for emotional problems should be developed for gay and bisexual individuals to help prevent sedative/hypnotic abuse.

Previous research has found that social support from peers and family is beneficial for the mental health of sexual minorities [17]. A previous study revealed that illegal substance use in sexual minority individuals was associated with family rejection, which indicates a lower degree of family support [18]. However, the manner in which family support interacts with this association still warrants exploration. Whether family support can moderate the association between homophobic bullying and sedative/hypnotic use in gay and bisexual individuals warrants further investigation. If family support can buffer the negative effects of homophobic bullying on sedative/hypnotic use, enhancing family support should be an essential part of prevention programs for sedative/hypnotic misuse in gay and bisexual individuals.

### 1.4. Aims of The Current Study

A previous study by the authors found that both homophobic bullying and low family support were significantly associated with sedative/hypnotic use among gay and bisexual men [7]. However, it remains unknown how emotional symptoms and family support influence this association. A comprehensive review proposed a conceptual moderated mediation model that illustrates the association between bullying and substance misuse, which is potentially mediated by internalized problems and stress, with family support as a moderator [19]. The present study aimed to develop a moderated mediation model to explain the association between homophobic bullying during childhood and adolescence and sedative/hypnotic use in early adulthood. According to the aforementioned studies, we supposed that current emotional symptoms mediated the association between homophobic bullying during childhood and adolescence and sedative/hypnotic use in early adulthood, while perceived family support during childhood and adolescence could moderate the mediating effect of emotional symptoms.

## 2. Materials and Methods

### 2.1. Participants and Procedure

The protocol of the current study was addressed in previously published studies [7,20]. In summary, gay and bisexual men aged 20 to 25 years were recruited through online and printed advertisements that were posted on social networking sites and within lesbian, gay, bisexual, and transgender clubs. Subjects who presented any cognitive decline that could interfere with their understanding of the purpose of the study or could affect their ability to complete the questionnaires were excluded.

The current cross-sectional study used a paper-and-pencil questionnaire. Research assistants explained the procedures and method for completing the questionnaires to each participant individually. Those who agreed to be recruited signed informed consent prior to completing the assessments. The participants could ask questions if they had any difficulty in completing the questionnaires, and the research assistants resolved any problems. The current study was approved by the Institutional Review Board of Kaohsiung Medical University Hospital, Kaohsiung, Taiwan (approval no. KMUHIRB-F(I)-20150026).

### 2.2. Measures

#### 2.2.1. Sedative/Hypnotic Use

An item on the Drug Use Disorders Identification Test-Extended (DUDIT-E) was used to determine the history of sedative/hypnotic use in the preceding month [21]. The concurrent validity of the D-score is reported to be acceptable, and its test–retest reliability is 0.79, indicating excellent intraclass correlation [21]. The responses for sedative/hypnotic use were graded on a six-point Likert scale as follows: 0—never used in the past year, 1—have used but less than once per month, 2—used once per month but less than twice per month, 3—used two to four times per month, 4—used two or three times per week, and 5—used four or more times per week.

#### 2.2.2. Homophobic Bullying

The six-item Chinese version of the self-reported School Bullying Experience Questionnaire (C-SBEQ) [22] was used to evaluate the participants’ experience of traditional homophobic bullying, including social exclusion, name calling, verbal abuse, physical abuse, forced work, and the confiscation of money, school supplies, or snacks at school, tutoring schools, after-school classes, or part-time workplaces. The participants were encouraged by the C-SBEQ to recall cases of the above experiences that occurred at elementary, junior high, and senior high schools. Previous experience of traditional homophobic bullying due to gender role nonconformity (6 items) and disclosure of sexual orientation (6 items) were assessed separately. The response for each item was graded on a four-point Likert scale as follows: 0—never, 1—just a little, 2—often, and 3—all the time. The total possible score ranged from 0 to 36. It has been previously reported that the C-SBEQ has acceptable reliability and validity [22]. Cronbach’s α values for the scales measuring the two types of bullying, one due to gender nonconformity and the other due to sexual orientation, were 0.79 and 0.82, respectively.

#### 2.2.3. Homophobic Cyberbullying

We used three items from the Cyberbullying Experiences Questionnaire (CEQ) [23] to assess participants’ experiences of cyberbullying, including any experiences of others posting mean or unpleasant comments, others posting upsetting pictures, photos, or videos, or online rumor-spreading through emails, blogs, social media platforms, and pictures or videos at the aforementioned school stages. Experiences of homophobic cyberbullying due to gender role nonconformity (3 items) and disclosure of sexual orientation (3 items) were assessed separately. The response for each item was graded on a four-point Likert scale that was the same as for the C-SBEQ, where the total score ranged from 0 to 18. Cronbach’s *α* values for the scales measuring cyberbullying due to gender nonconformity and sexual orientation were 0.71 and 0.86, respectively.

#### 2.2.4. Depressive Symptoms

The present study evaluated the severity of depressive and anxiety symptoms. The 20-item self-administered Mandarin Chinese version of the Center for Epidemiological Studies-Depression Scale (MC-CES-D) was used to estimate the frequency of depressive symptoms in the week preceding the study [24,25]. The total score could range from 0 to 60, and the items were graded on a four-point Likert scale as follows: 0—never or less than one day per week, 1—one or two days per week, 2—three or four days per week, and 3—five to seven days per week. A higher total score indicated more severe depression. Cronbach’s *α* value for the MC-CES-D in the current study was 0.92.

#### 2.2.5. Anxiety Symptoms

A total of 20 items from the self-reported State-Trait Anxiety Inventory (STAI-S) questionnaire Y were used to evaluate the individuals’ current anxiety symptoms [26,27]. The items were graded on a four-point Likert scale, with total scores ranging from 20 to 80. The grades were defined as follows: 1—never, 2—just a little, 3—often, and 4—all the time. A higher total STAI-S score represented more severe anxiety. Cronbach’s *α* value for the STAI-S in the present study was 0.87.

#### 2.2.6. Family Support

The Chinese version of the 5-item self-administered Family Adaptation, Partnership, Growth, Affection, Resolve (APGAR) Index was used to measure the participants’ satisfaction with family support during their childhood and adolescence [28,29]. Each item was rated on a four-point Likert scale as follows: 0—never, 1—just a little, 2—often, and 3—all the time. The total possible score ranged from 0 to 15. A higher total score on the Family APGAR Index represented higher levels of family support. The Cronbach *α* value for the Family APGAR Index in the current study was 0.86. In the following analysis, total scores of Family APGAR were used as the “family support” factor.

#### 2.2.7. Statistical Analysis

SPSS version 23.0 for Windows (SPSS Inc., Chicago, IL, USA) was used to perform the statistical analysis. Descriptive analysis including mean ± standard deviation (SD) were used for continuous variables. Normality of variables were estimated using the Kolmogorov–Smirnov test, and the correlation matrix across all variables was developed by Pearson or Spearman correlation. Exploratory factor analysis (EFA) was used for dimension reduction. The total scores of C-SBEQ and CEQ were transformed into the “bullying victimization” factor, and the total scores of the MC-CES-D and the STAI-S were transformed into the “emotional symptoms” factor to fit our conceptual model (Figure 1). The authors hypothesized that the association between “bullying and victimization” and “sedative/hypnotic use” was mediated by “emotional symptoms”, and the magnitude of the indirect effect (mediation) was moderated by “family support”. To test the moderated indirect effect, the mediation models were estimated and the moderated mediation models were created using PROCESS macro version 3.4 developed by Hayes [30,31] based on the hypothesis. In PROCESS, different numbers are used to specify different preset complex models and Model 14 was applied to the moderated mediations in the conceptual model shown in Figure 1. PROCESS performs ordinary least squares regression to estimate the moderated indirect effect. In the moderated mediation analysis, all of the quantitative variables were centralized [32], and the 95% percentile bootstrap confidence interval (CI) with 5000 bootstrapping samples was calculated. The index of moderated mediation and its 95% CI, as estimated by PROCESS, were used to determine and quantify the statistical significance of the moderated mediation effect [30]. If the 95% CI did not include zero, it meant that the moderated mediation effect was statistically significant. If the moderated mediation effect was statistically significant, the conditional indirect effects of bullying on sedative/hypnotic use (through emotional symptoms) were evaluated at three different levels of family support, corresponding to the values of mean plus SD, mean, and mean minus SD.

## 3. Results

### 3.1. Description of Patient Variables

In total, 500 males (129 bisexual men and 371 gay men) were recruited, who had a mean age at 22.94 years. The summaries of variables, normality test, and collinearity matrix are listed in Table 1.

Since the significant Kolmogorov–Smirnov test (*p* < 0.05) indicated the non-normal distribution of all variables, the bootstrapping method with 5000 samples was suitable for estimation. Before entering the analysis for moderated mediation, EFA was used for dimension reduction, and estimates of the factor loading are listed in Table 2.

### 3.2. Tests For The Moderated Mediation Model

The moderated mediation analysis was conducted using the factor scores after dimension reduction. The results of ordinary least squares regression are summarized in Table 3 and illustrated in Figure 2, with estimates of patch and significance (marked as ***, *p* < 0.001). Bullying was positively associated with the severity of emotional symptoms (a pathway with β = 0.29, *p* < 0.001). The severity of emotional symptoms was also positively associated with sedative/hypnotic use (b pathway with β = 0.19, *p* < 0.001). The index of moderated mediation was estimated to be 0.14 with a 95% CI of −0.0253 to −0.0048, indicating a significantly negative moderation effect. In summary, these results supported the positive indirect effect of bullying and victimization on sedative/hypnotic use through the positive mediating effect of emotional symptoms. A lower level of family support was shown to enhance the effect of emotional symptoms, thus revealing a negative moderating effect.

To better understand the moderating effect of family support, the bootstrap indirect effects were estimated for the mediating effect of the severity of emotional symptoms at three different levels of family support (mean plus SD, mean, and mean minus SD). The 95% CI shows that the two higher values of family support did not contain zero, whereas the two lower values did (Table 4); this is illustrated in Figure 3. To sum up, the moderating effect of family support was confirmed, as the mediating effect of emotional symptoms in the association between bullying and sedative/hypnotic use was weaker at higher levels of exposure, thus indicating the protective effect of family support.

## 4. Discussion

It was found that a higher level of homophobic bullying was significantly associated with sedative/hypnotic use among gay and bisexual men, and this was mediated by a higher severity of emotional symptoms. Furthermore, this pathway was weakened by the moderating effect of family support on the association between emotional symptoms and sedative/hypnotic use. This indicated that a higher level of family support protected gay and bisexual men who had experienced homophobic bullying in their childhood and adolescence from later sedative/hypnotic use.

### 4.1. Mediating The Effect of Emotional Symptoms

Several studies have previously addressed the mediating effect of depression or anxiety on the association between childhood bullying and substance use. A study recruiting African adolescents revealed a predominant indirect pathway between bullying and tobacco and alcohol use through the mediation of loneliness [33]. Another paper reported that the effect of peer victimization on alcohol use was mediated by depression and anxiety among white youths [34]. Two previous studies have indicated that depressive symptoms mediate the effect of bullying on substance use in females but not in males [35,36]. However, these two previous studies did not record the sexual orientation of participants and they did not examine the mediating model for various substances.

The result of the present study supports the mediating effect of emotional symptoms on the association between homophobic bullying in childhood and adolescence to sedative/hypnotic use in early adulthood. It indicated that if gay and bisexual men experienced homophobic bullying and symptoms of depression or anxiety emerged, they may easily suffer from the abuse of sedative/hypnotic drugs. Hence, timely intervention for their depression or anxiety may help prevent the later sedative/hypnotic use. Further investigation is needed to replicate the results of the present study in lesbians and those using substances other than sedatives/hypnotics.

### 4.2. Moderating The Effect of Family Support

Previous research has reported that low family support was significantly associated with substance use [37], and that cyberbullying was positively correlated with internalized mental health problems (anxiety, depression, self-harm, suicide ideation, and suicide attempts), substance use problems, and few family contacts [38]. A female-based study also demonstrated that lower parent–child communication predicted substance use for sexual minority young females [39]. However, the moderating effect of family support on the association between bullying and substance use is seldom discussed. The current study further extended the application of the previously purposed model [19].

A higher level of family support during childhood and adolescence was revealed to have a protective effect for gay and bisexual men who experienced homophobic bullying and suffered from emotional problems, and decreased the possibility for later sedative/hypnotic abuse. Hence, timely education and support to create a LGBT-friendly environment for family who have gay or bisexual youths can weaken the effect of emotional symptoms on sedative/hypnotic abuse. Early intervention to enhance family support for gay and bisexual youths is crucial to enhance mental health.

### 4.3. Limitations

Several limitations in this study must be highlighted. First, the details of sedative/hypnotic use were not recorded, such as the source of the sedatives/hypnotics, and therefore it was not possible to determine whether the sedatives/hypnotics that the participants used were prescribed by a doctor. Second, as it was a questionnaire survey study, it could not determine diagnoses of sedative/hypnotic abuse disorders or other psychiatric comorbidities. Third, this study only recruited gay and bisexual men, so the generalizability of the findings may be limited. Fourth, the lack of the other relevant variables that could contribute to explaining the conceptual model is another limitation. Finally, this self-reported study retrospectively acquired information on homophobic bullying and family support; therefore, recall bias may exist.

## 5. Conclusions

The current study identified that a higher degree of homophobic bullying was positively associated with sedative/hypnotic use among gay and bisexual men, and this was mediated by emotional symptoms. Family support played a protective role for weakening the association between emotional symptoms and sedative/hypnotic use. The study also demonstrated that, to help prevent the onset of sedative/hypnotic use, mental health professionals should make an effort to treat the emotional symptoms of gay and bisexual men who have had previous experiences of homophobic bullying. Moreover, early identification of homophobic bullying during childhood and adolescence is also crucial. The establishment of an LBGT-friendly environment in school, regular screening of mental health status, and timely intervention are beneficial for gay and bisexual men to prevent the emergence of mental health problems in the future. Furthermore, the present study demonstrated the importance of developing early intervention programs for gay and bisexual men who have poor family support. LGBT health education, support of associated resources, and peer groups for family members of sexual minorities may help parents to develop a friendly family relationship with sexual minority youths. Further research on unexplored factors can help us enrich the conceptual model built by the current studies. Studies regarding gender role conformity, internal stigmatization, and assessment of other supporting systems will be beneficial to clarifying the substance use problems occurring among gay and bisexual men.

## Figures and Tables

**Figure 1 ijerph-17-03870-f001:**
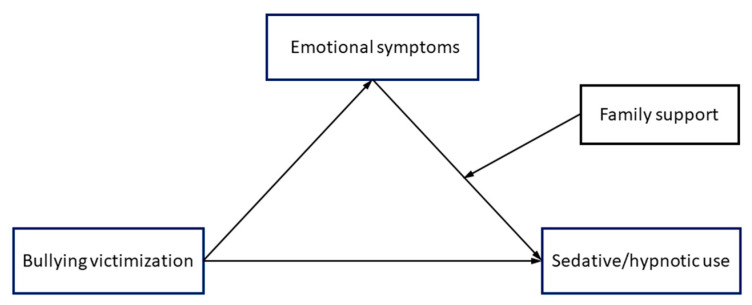
Conceptual model of the moderated mediation.

**Figure 2 ijerph-17-03870-f002:**
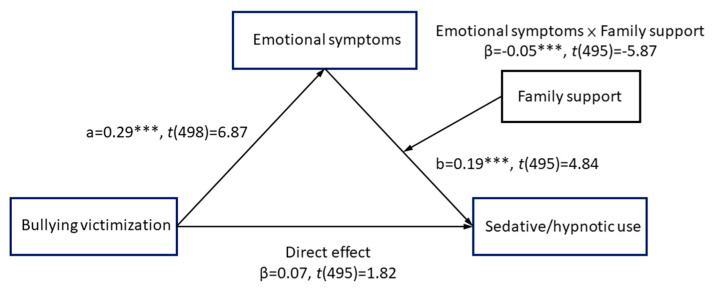
Final model indicating the co-efficient estimates and statistical significance.

**Figure 3 ijerph-17-03870-f003:**
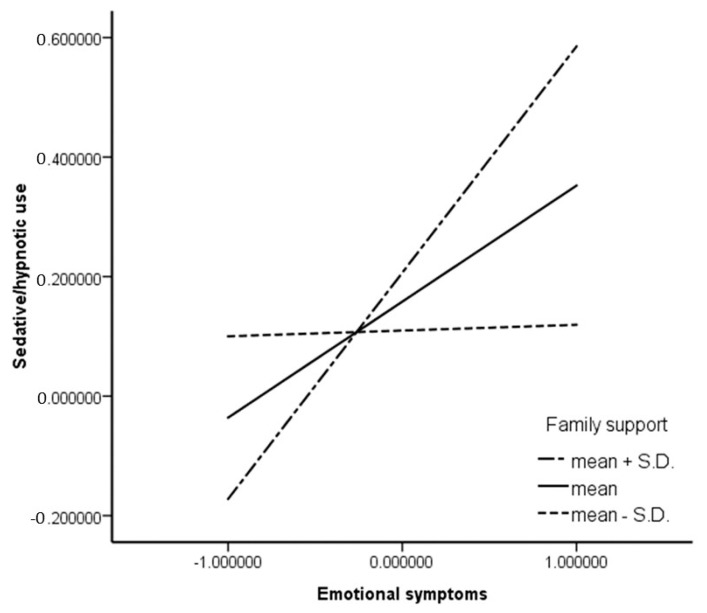
Distribution plots for three different effects of moderation.

**Table 1 ijerph-17-03870-t001:** Distribution, normality estimates, and correlation matrix for all quantitative variables (*n* = 500).

Variables	Mean (SD)	Kurtosis (Skewness)	1	2	3	4	5	6	7
1. Age	22.94 (1.57)	−0.21 (−1.15)	-	0.05	−0.06	0.03	0.01	0.04	0.01
2. Sedative/hypnotic use	0.2 (0.87)	4.27 (18.01)		-	−0.18 *	0.20 *	0.16 *	0.27 *	0.21 *
3. Family support	8.49 (3.83)	−0.30 (−0.62)			-	−0.28 *	−0.17 *	−0.39 *	−0.33
4. Traditional bullying	5.01 (5.03)	1.85 (4.74)				-	0.53 *	0.32 *	0.25 *
5. Cyberbullying	1.28 (2.29)	2.80 (10.84)					-	0.24 *	0.17 *
6. Depression	17.47 (10.32)	0.79 (0.34)						-	0.72 *
7. Anxiety	20.40 (11.60)	0.37 (−0.38)							-

SD—Standard deviation; * = *p* < 0.05.

**Table 2 ijerph-17-03870-t002:** Factor loading scores estimated by exploratory factor analysis for dimension reduction.

Variables	Factor 1	Factor 2
Depression	0.929	-
Anxiety	0.929	-
Traditional bullying victimization	-	0.881
Cyberbullying victimization	-	0.881

Factor 1—Emotional symptoms; Factor 2—Bullying victimization.

**Table 3 ijerph-17-03870-t003:** Ordinary least squares regression results for moderated indirect effect.

**Outcome Variable: Emotional Symptoms ^1^**					
Predictors	β	SE	*p*	LLCI	ULCI
Bullying victimization	0.294	0.043	<0.001	0.210	0.378
**Outcome variable: Sedative/Hypnotic Use ^2^**					
Predictors	β	SE	*p*	LLCI	ULCI
Bullying victimization	0.070	1.824	0.069	−0.005	0.145
Emotional symptoms	0.194	4.844	<0.001	0.115	0.273
Family support	−0.013	−1.228	0.220	−0.033	0.008
Emotional symptoms × family support	−0.048	−5.867	<0.001	−0.064	−0.032
Index of moderated mediation	β	SE	*p*	LLCI	ULCI
Family support	−0.0142	0.0052	-	−0.0253	−0.0048

LLCI: lower limit of 95% confidence interval; ULCI: upper limit of 95% confidence interval; β: regression coefficient; SE: standard error; ^1^: Model F(1, 498) = 47.20; *p* < 0.001; ^2^: Model F(4, 495) = 25.10; *p* < 0.001.

**Table 4 ijerph-17-03870-t004:** Moderated indirect effect of bullying victimization on sedative/hypnotic use divided into three levels of family support.

Family Support	Indirect Effect	Bootstrap SE	95% of CI ^1^
Mean—SD (4.6561)	0.1114	0.0342	(0.0513, 0.1853)
Mean (8.488)	0.0571	0.0173	(0.0283, 0.0966)
Mean + SD (12.3199)	0.0028	0.0152	(−0.0265, 0.0336)

^1^—percentile bootstrap confidence interval.

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
