# Peer review of "The Impact of Emotional Symptoms and Family Support on the Association Between Homophobic Bullying and Sedative/Hypnotic Use among Gay and Bisexual Men in Taiwan: A Moderated Mediation Model"

_ijerph, 2020, doi:10.3390/ijerph17113870_

Round 1
Reviewer 1 Report
The manuscript entitled "The impact of emotional symptoms and family support on the association between homophobic bullying and sedative/hypnotic use among gay and bisexual men in Taiwan: A moderated mediation model", investigates the mediating effect of emotional symptoms and the moderating effect of family support on the relationships between sedative/hypnotic use and homophobic bullying in gay and bisexual male participants.
I think that the manuscript could be of really interest for the readers of IJERPH. It covers a relevant research topic. The results are clear and the methodology is strong. Also, I appreciated the updated bibliography.
However, I think that moderate revisions should be necessary before a possible publication. The main aspects I would like the authors to focus, are related to the introduction and the discussion sections.
Below you can find my suggestions to improve the manuscript:
- The first paragraph of the introduction titled " Sedative/hypnotic use in gay and bisexual men" should be deepened. I would suggest to focus more on the gay and bisexual population. In addition, no mention to the specific national context of the study is present.
- Rather than saying that no previous studies investigated the relationships you explored, you should stress the relevance of investigating them. Why is this important? What are the theoretical and practical aspects that your research add to the field?
- I should argue more the reasons why the authors choose these specific variables as moderator and mediator. Also, there are other variables which would have enriched their model (i.e. internalized sexual stigma, adherence to traditional gender roles, and others). I think the authors should at least mention these variables in their introduction and provide some references to support their implication on the phenomena of bullying and substance use and other mental problems in sexual minority people. Indeed, tot surprisingly, your dependent measures of bullying and cyberbullying have specific items and subscales referring to the violation of traditional gender roles.
- Regarding the method, I did not understand how the authors check the participants who presented any cognitive decline. Please, specify it better.
- In the results section I would integrate the first paragraph about descriptives, by adding the analysis of normality and multicollinearity. I would suggest the authors to add a correlation table in which they can add means, SDs, kurtosis and skewness values of all variables. (The table 1 would be redundant).
- I think the discussion section would benefit of substantial revision. Several parts should be better in the introduction: lines 233-240; lines 248-252. Here, I would stress more the relevance and the implication of your results. It is not necessary to divide your discussion section in more subsections, in relation to moderating and mediating effects.
- The first sentence of limitations is redundant and I think it could be removed.
- In the limitations I would mention the lack of the other relevant variables which could contribute to explain the relationships investigated.
- I would suggest to add a paragraph named "Further research directions", or it could be integrated in the limitations section, and the authors might argue about this.
I might suggest some relevant references which could help the authors to improve their introduction and discussion, and which could be added to the bibliography:
-Salvati, M., Pistella, J., & Baiocco, R. (2018). Gender roles and internalized sexual stigma in gay and lesbian persons: A quadratic relation. International Journal of Sexual Health, 30(1), 42-48.
- Baiocco, R., Pistella, J., Salvati, M., Ioverno, S., & Lucidi, F. (2018). Sports as a risk environment: Homophobia and bullying in a sample of gay and heterosexual men. Journal of Gay & Lesbian Mental Health, 22(4), 385-411.
I hope my suggestions could help the authors to improve their manuscript.
Best regards.
Author Response
as attached file

Reviewer 2 Report
Dear authors,
First of all I would like to congratulate you for your work. I really find your topic very appealing and original. But I think you should improve your introduction and conclusion parts. Regarding to Introduction you mention a study from the USA and I don't really know if that could be connected and applied to Taiwan. About the conclusion I find it too short overall if we take under consideration all the interesting information you found.
Very Best
Author Response
as attached file
